# Deletion of podocyte Rho-associated, coiled-coil-containing protein kinase 2 protects mice from focal segmental glomerulosclerosis
Keiichiro Matoba [1] ✉, Yosuke Nagai [1], Kensuke Sekiguchi[1], Shinji Ohashi[1], Etsuko Mitsuyoshi[1], Masayuki Shimoda[2], Toshiaki Tachibana[3], Daiji Kawanami [4], Tamotsu Yokota[1], Kazunori Utsunomiya[5] & Rimei Nishimura[1]

Focal segmental glomerulosclerosis (FSGS) shares podocyte damage as an essential pathological finding. Several mechanisms underlying podocyte injury have been proposed, but many important questions remain. Rho-associated, coiled-coil-containing protein kinase 2 (ROCK2) is a serine/threonine kinase responsible for a wide array of cellular functions. We found that ROCK2 is activated in podocytes of adriamycin (ADR)-induced FSGS mice and cultured podocytes stimulated with ADR. Conditional knockout mice in which the ROCK2 gene was selectively disrupted in podocytes (PR2KO) were resistant to albuminuria, glomerular sclerosis, and podocyte damage induced by ADR injection. In addition, pharmacological intervention for ROCK2 significantly ameliorated podocyte loss and kidney sclerosis in a murine model of FSGS by abrogating profibrotic factors. RNA sequencing of podocytes treated with a ROCK2 inhibitor proved that ROCK2 is a cyclic nucleotide signaling pathway regulator. Our study highlights the potential utility of ROCK2 inhibition as a therapeutic option for FSGS.

Focal segmental glomerulosclerosis (FSGS) is a histologic pattern of glomerular injury with heterogenic causes. Experimental studies have illuminated podocyte injury as a common feature of FSGS that eventually leads to effacement of the foot processes from the surface of the capillaries and impaired glomerular filtration in which larger molecules, such as proteins, pass into the urine. Despite the global health burden, current pharmacological therapies cannot fully prevent or treat podocyte damage, and a number of patients with FSGS still progress toward kidney failure. Achieving a thorough understanding of podocyte biology and kidney care provision is thus becoming increasingly important.

Rho-associated, coiled-coil-containing protein kinase (ROCK) is a ubiquitously expressed serine/threonine kinase that governs a wide range of physiologic or pathologic responses that vary based on the cell lineage and upstream stimulus. For example, ROCK is a regulator of cell migration and motility through modulation of cytoskeletal rearrangement. Recent studies have shown that kidney ROCK activity is elevated in both animal models of diabetes and patients with diabetes[1,2]. Furthermore, the pharmacological suppression of ROCK can prevent or treat diabetic kidney damage by attenuating inflammation and hypoxic reactions[2–4].

ROCK has two distinct isoforms: ROCK1 and ROCK2. Findings in isoform-specific gene deletion mice argue for the roles of ROCK1 and ROCK2 in the organization of actomyosin bundles and blood coagulation, respectively[5,6]. With respect to the kidney, we previously demonstrated that ROCK1 deletion is protective maintaining the glomerular function by promoting AMP-activated protein kinase-regulated metabolism[7]. In addition, hyperglycemia alters podocyte homeostasis by inducing dynamin-related protein 1-mediated mitochondrial fission through ROCK1, which leads to an abnormal kidney function[8]. In contrast, ROCK2 inhibits peroxisome proliferator-activated receptor α, thereby regulating fatty acid utilization[9]. Furthermore, ROCK2 has been shown to mediate vascular

¹Division of Diabetes, Metabolism and Endocrinology, Department of Internal Medicine, The Jikei University School of Medicine, Tokyo 105-8461, Japan. ²Department of Pathology, The Jikei University School of Medicine, Tokyo 105-8461, Japan. ³Core Research Facilities for Basic Science, Research Center for Medical Science, The Jikei University School of Medicine, Tokyo 105-8461, Japan. ⁴Department of Endocrinology and Diabetes, Fukuoka University School of Medicine, Fukuoka 814-0180, Japan. ⁵Nomura Hospital, Tokyo 181-8503, Japan. ✉e-mail: matoba@jikei.ac.jp

**Fig. 1 | ROCK2 is upregulated in FSGS and in ADR-stimulated podocytes.** Renal ROCK2 distribution in human (**a**) and mouse (**b**). Single-cell RNA sequencing data were acquired from the Kidney Interactive Transcriptomics database (https://humphreyslab.com/SingleCell). Pod, podocyte; MC, mesangium; EC, endothelial cells; PT, proximal tubules; LOH (AL), loop of Henle, ascending limb; DCT, distal convoluted tubules; CNT, connecting tubules; PC, principal cells; IC-A, type A intercalated cells; IC-B, type B intercalated cells; tSNE, t-distributed stochastic neighbor embedding. **c** A correlation analysis between glomerular ROCK2 mRNA and urinary protein excretion in patients with FSGS. Hodgin FSGS Glom data obtained from the transcriptomic database Nephroseq (https://www.nephroseq.com) were used. The trendline is shown in red. The R-value was determined by Pearson correlation analysis (*n* = 6). **d** ROCK2 protein abundance in the renal cortex of ADR-injected mice (*n* = 3). **e** Representative immunostaining of ROCK2 (green) and nephrin (red) in the glomerulus of ADR-injected mice. Nuclei were visualized with DAPI (blue). The scale bar on the top left represents 10 μm. **f** The transcript levels of ROCK2 in glomeruli isolated from ADR-injected mice (*n* = 3). **g** The ROCK2 mRNA levels after ADR stimulation in cultured podocytes (*n* = 4). **h** ROCK2 protein levels in ADR-treated podocytes (*n* = 3). *$p < 0.05$. Data represent the mean ± s.e.m.

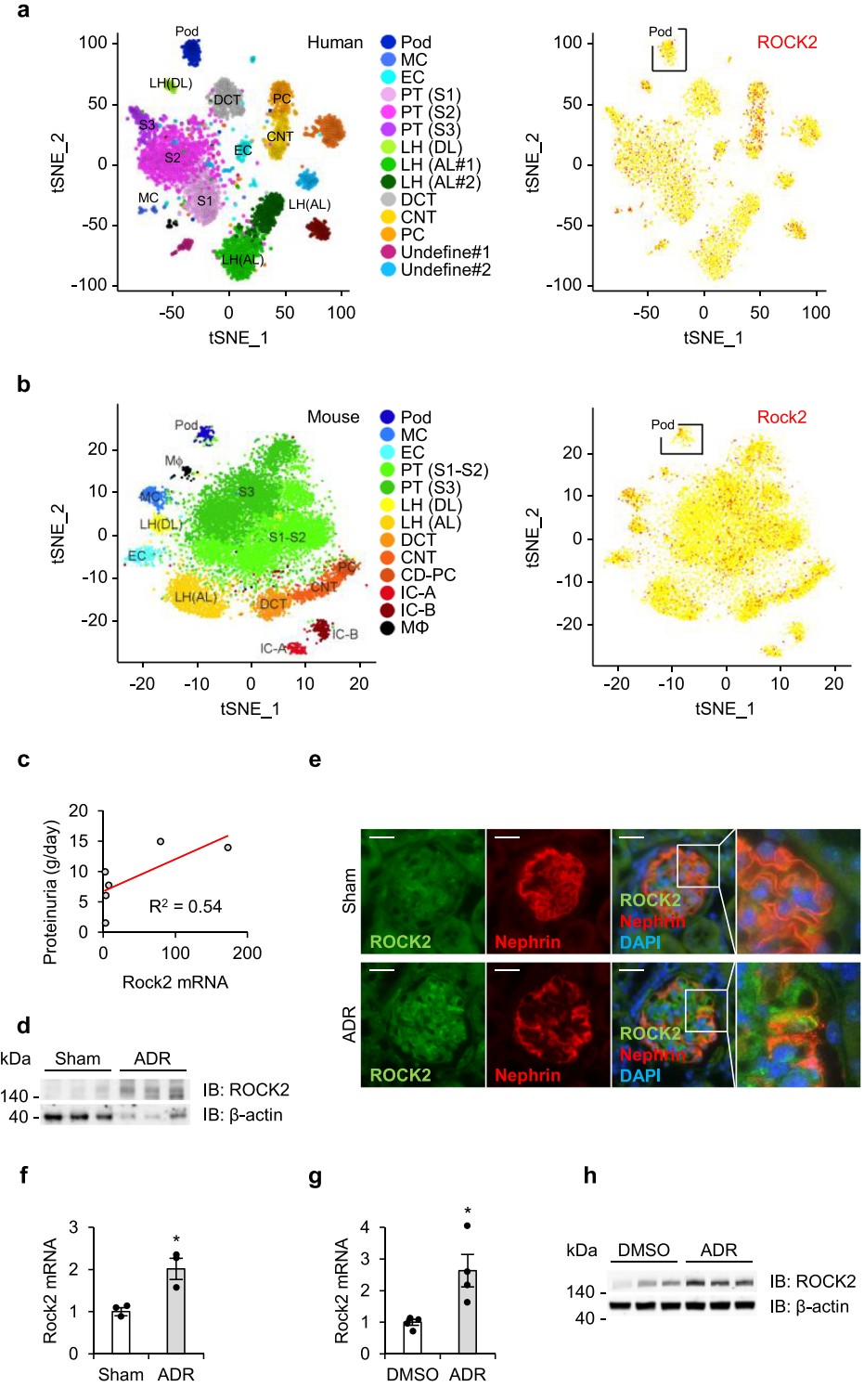

inflammation[10]. However, the role of ROCK2 in FSGS attributed to podocyte damage remains unclear.

To address this question and in turn uncover potential therapeutic targets against FSGS, we investigate the functional aspects of podocyte ROCK2 in vivo and in vitro. ROCK2 is upregulated in ADR-induced glomerular injury, a rodent model of FSGS characterized by podocyte damage followed by glomerulosclerosis, tubulointerstitial inflammation, and fibrosis. We show that ablation of ROCK2 in podocytes prevents podocyte damage and albuminuria induced by ADR injection. In addition, we performed pharmacological intervention for ROCK2 to demonstrate protection against podocyte loss and glomerulosclerosis in ADR-injected mice.

## Results

### Induction of ROCK2 expression in FSGS

To gain insight into the potential role of ROCK2 in FSGS, we first assessed its distribution in kidney cell types using the kidney single-cell RNA sequencing database, Kidney Interactive Transcriptomics (https://humphreyslab.com/SingleCell). As demonstrated in Fig. 1a and b, ROCK2 was distributed

in various types of human and murine kidney cells, including podocytes, mesangial cells, endothelial cells, and tubules.

We next examined the ROCK2 expression in patients with FSGS and in ADR-induced FSGS mice. As shown in Fig. 1c, transcript datasets obtained from Nephroseq (https://www.nephroseq.org) demonstrated a positive correlation between the glomerular expression of ROCK2 and urinary protein excretion in patients with FSGS, which suggests the association of activated ROCK2 signaling with an abnormal kidney function. In addition, kidney ROCK2 protein levels were elevated in ADR-injected mice compared to vehicle-injected sham mice (Fig. 1d).

We next performed immunohistochemistry on kidney sections of ADR-injected mice to determine the distribution of ROCK2. In ADR-injected mice, ROCK2 was strongly detected in glomerular cells including podocytes (Fig. 1e). Furthermore, glomeruli obtained from these mice exhibited a significant elevation in transcript levels of ROCK2 compared with samples isolated from sham mice (Fig. 1f). We then confirmed the expression of ROCK2 in cultured podocytes stimulated with ADR. The expression of ROCK2 was significantly induced at the mRNA level after ADR exposure (Fig. 1g). Consistent with the transcript levels of ROCK2, its protein expression was elevated following ADR treatment (Fig. 1h). Taken together, these results support the notion that ROCK2 expression is augmented during ADR-induced podocyte injury.

## Effects of podocyte-specific ROCK2 deletion on glomerular integrity and progression of kidney fibrosis in mice

To begin testing for functional links between ROCK2 expression and ADR-induced kidney damage, we crossed Rock2-flox mice with mice expressing Cre recombinase under the control of the Nphs2/Podocin promoter to generate mice with podocyte-specific disruption of ROCK2. Podocyte-specific ROCK2 ablation was confirmed by immunohistochemistry (Supplementary Fig. 1). For our experiments, we used Nphs/Podocin-Cre$^+$ Rock2$^{fl/fl}$ mice (PR2KO) and Nphs/Podocin-Cre$^-$ Rock2$^{fl/fl}$ (WT) littermates. PR2KO mice were viable, fertile, and without discernable defects in phenotype. To determine the function of podocyte ROCK2 in FSGS, WT and PR2KO mice were administered ADR intravenously (20 mg/kg) 4 weeks before morphological and biochemical analyses (Fig. 2a). Saline-injected mice served as controls. In the setting of ADR nephropathy, the serum creatinine levels tended to be reduced in PR2KO mice compared to WT mice, but this difference was not statistically significant (Fig. 2b). However, the selective ablation of ROCK2 in podocytes resulted in significant prevention of the increase in urinary albumin excretion (Fig. 2c). Kidney sections were then analyzed for histological injuries typically observed in FSGS. In contrast to WT mice, those lacking ROCK2 exclusively in podocytes were completely impervious to glomerular sclerosis (Fig. 2d).

We next counted the number of podocytes by investigating WT1-positive cells in glomeruli. As demonstrated in Fig. 2e, healthy glomeruli in sham mice showed uninterrupted WT1 staining. While a significant reduction in podocytes was observed in ADR-injected WT mice, PR2KO mice were protected from the loss of podocytes. The ultrastructure of glomeruli was visualized by TEM. As shown in Fig. 2f and g, TEM micrographs of ADR-injected mice showed widespread foot process effacement and diffuse thickening of GBM. In contrast, PR2KO mice demonstrated a significant reduction in foot process width and GBM thickness in the setting of FSGS.

Because urinary albumin can be reabsorbed in the tubules, leading to tubulointerstitial injury, we investigated whether or not the ablation of ROCK2 in podocytes influenced damage in the tubular compartment. To this end, we quantified the area of interstitial fibrosis in WT mice and PR2KO mice subjected to ADR. As in the glomerulus, tubulointerstitial fibrosis was attenuated in PR2KO mice (Fig. 2h). To discern the molecular mechanisms underlying these beneficial effects of ROCK2 inhibition, we analyzed the gene expression of profibrotic genes in kidney tissues. Concordant with the functional and microscopic data, quantitative PCR of kidney tissue identified the upregulation of Tgfb1, Pai1, Acta2, and Fn1,

which encode transforming growth factor β-1, plasminogen activator inhibitor-1, α-smooth muscle actin, and fibronectin-1, respectively. These inductions were significantly prevented in the kidneys of PR2KO mice (Fig. 2i).

Taken together, these findings show that specific deficiency of ROCK2 rescued podocyte damage, and as a consequence, glomeruli and interstitial lesions were protected from sclerotic changes.

## Effects of pharmacological intervention for ROCK2 on kidney damage in ADR-injected mice

As the above results establish the importance of ROCK2 in the progression of FSGS, we next sought to determine the therapeutic effects of ROCK2 blockade.

ADR-injected mice were treated with either SLx-2119, a ROCK2 inhibitor, or vehicle for 2 weeks (Fig. 3a). Treatment with SLx-2119 significantly ameliorated ADR-induced albuminuria compared to vehicle treatment without affecting serum creatinine levels (Fig. 3b, c). Furthermore, glomerular fibrosis was significantly reduced in SLx-2119-administered ADR mice compared with vehicle-treated mice (Fig. 3d). Consistent with the results obtained from PR2KO mice, the loss of glomerular podocytes was attenuated in mice treated with SLx-2119 (Fig. 3e). The ultrastructures of podocytes (Fig. 3f) and GBM (Fig. 3g) were also protected from ADR injury in these mice. As demonstrated in Fig. 3h, tubulointerstitial fibrosis was significantly inhibited by the pharmacological intervention for ROCK2.

To analyze the downstream effects of podocyte ROCK2 on fibrosis mediators, we measured the mRNA expression of a panel of factors relevant to kidney fibrosis by quantitative PCR (Fig. 3i). We found that SLx-2119 treatment significantly suppressed the induction of profibrotic genes in the kidney (Fig. 3i). Taken together, these findings indicate that pharmacological ROCK2 inhibition can attenuate kidney histological and functional damage by, at least in part, suppressing fibrotic regulators in the ADR-induced mouse model.

## Role of ROCK2 in cyclic nucleotide signaling pathways

As a complementary approach, we next investigated the molecular basis by which ROCK2 blockade provides protective actions on podocytes.

As shown in Fig. 4a, transcriptomic profiles were assessed in ADR-stimulated podocytes. The criteria for the detection of differentially expressed genes in this experiment were an absolute fold-change value of > 2 and high statistical significance ($p$ value < 0.05). We found that 412 mRNAs were significantly upregulated, while 239 mRNAs were downregulated in ROCK2-inhibited podocytes (Fig. 4b, c). When a gene enrichment analysis was used to identify key pathways driving these transcriptional changes, the top differentially expressed pathways in these cells included the cyclic GMP (cGMP)-dependent protein kinase (PKG) and cyclic AMP (cAMP) signaling pathways (Fig. 4d). The upregulated genes involved in cyclic nucleotide signaling included the gene encoding regulator of G protein signaling 2 (RGS2), which prevents the progression of kidney fibrosis[11]. Induction of RGS2 was confirmed by quantitative PCR in podocytes treated with ROCK2 inhibitor (Fig. 4e), siRNA against ROCK2 (Fig. 4f), and primary podocytes obtained from PR2KO mice (Fig. 4g). The protein levels of RGS2 were analyzed using ELISA. The results in Fig. 4h demonstrate that RGS2 protein levels increased with SLx treatment under ADR-stimulated conditions. However, kidney tissue samples did not yield similar results (Fig. 4i). These findings suggest that ROCK2-mediated regulation of RGS2 may be restricted to podocytes. To assess the role of RGS2 in mediating ROCK2's function, we next conducted a TUNEL assay under ADR-stimulated conditions. As demonstrated in Fig. 4j, the number of TUNEL-positive cells was decreased by the treatment of SLx. However, the protective action of ROCK2 inhibition was partially canceled by the treatment of siRNA against RGS2. These data support the idea that the beneficial actions of ROCK2 inhibition are dependent, at least in part, on RGS2 upregulation. Of the molecules selectively regulated by ROCK2, genome-scale integrated analysis of gene networks in tissues (GIANT) indicated that cadherin 13

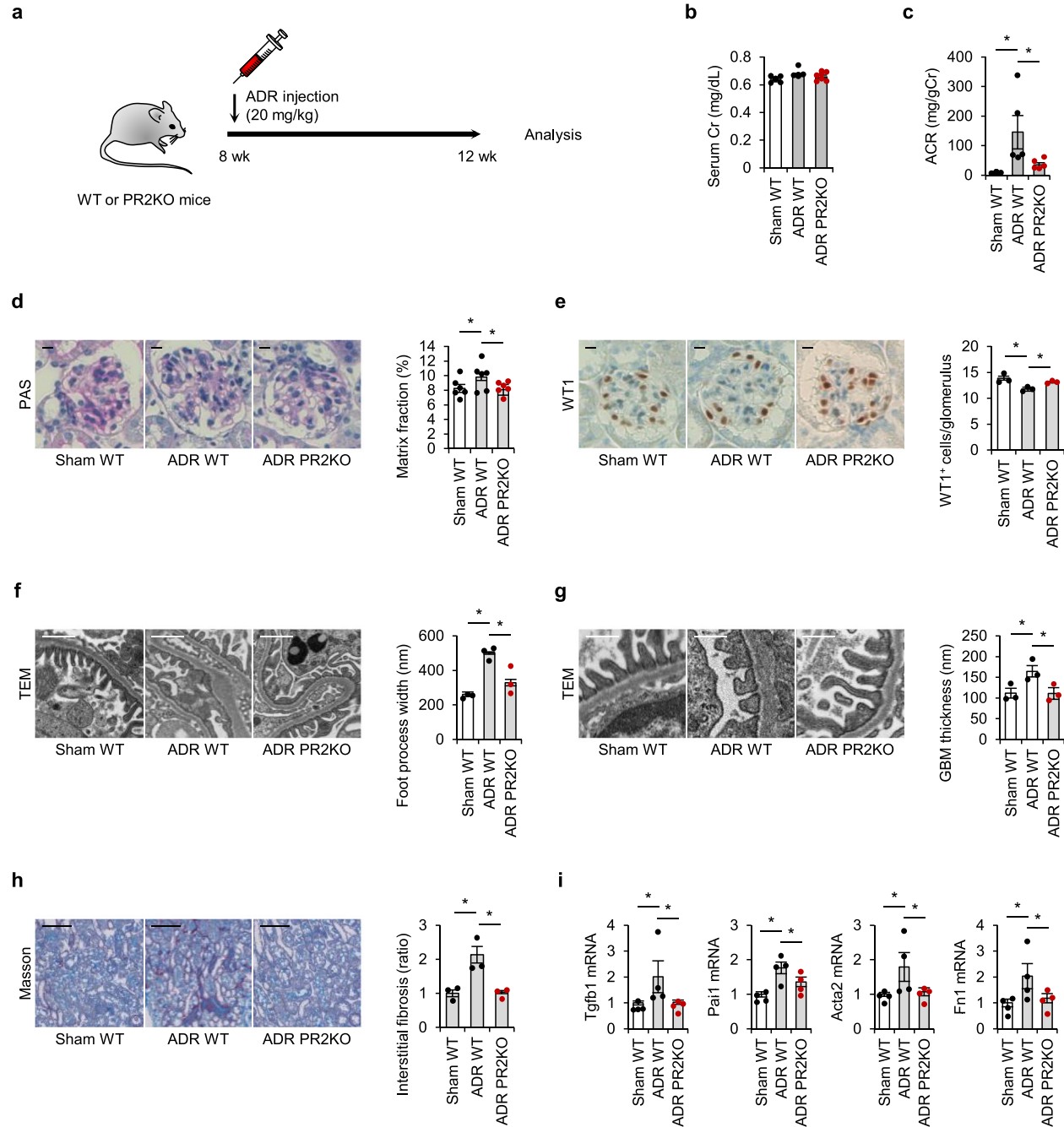

**Fig. 2 | Specific ablation of podocyte ROCK2 prevents FSGS progression in mice.**
**a** The scheme of the strategy for the generation of ADR mice. **b** Serum creatinine (Cr) levels at 12 weeks of age ($n = 6, 7$). **c** Urinary albumin to creatinine ratio (ACR) in ADR-injected mice ($n = 7$). **d** Representative periodic acid-Schiff (PAS)-stained images of kidney glomeruli from mice. The scale bar on the top left represents 10 μm ($n = 6$). **e** Wilms tumor 1 (WT1) immunostaining and quantification of WT1-positive cells in glomeruli from mice. The scale bar on the top left represents 10 μm ($n = 3$). **f** Foot process width examined by transmission electron microscopy (TEM). The scale bar on the top left represents 1 μm ($n = 3$). **g** GBM thickness assessed by TEM. The scale bar on the top left represents 0.5 μm ($n = 3$). **h** Masson's trichrome-stained kidney sections. The scale bar on the top left represents 100 μm. Fibrosis volume was quantified as the area stained positive for collagen. **i** The expression levels of fibrosis mediators in the kidney ($n = 4$). *$p < 0.05$. Data represent the mean ± s.e.m.

(CDH13) provided the strongest predicted functional connection between ROCK2 and RGS2 selectively in podocytes (Fig. 4g).

## Discussion

Podocytes are highly specialized epithelial cells that cover the outer layer of the GBM. Since the podocyte loss is a major determinant of both experimental and human FSGS[12,13], the identification of potential therapeutic targets for preventing podocyte damage has clinical importance for the

treatment of FSGS. Using genetic and biochemical approaches, we identified ROCK2 as an important mediator of albuminuria in a murine model of FSGS. We have now provided proof of concept that ROCK2 inhibition attenuated glomerular sclerosis, GBM abnormalities, and subsequent tubulointerstitial fibrosis.

In the present study, we showed that podocyte ROCK2, the expression of which is increased under ADR-stimulated conditions, is a key mediator of histological and functional abnormalities in FSGS. In earlier loss-of-

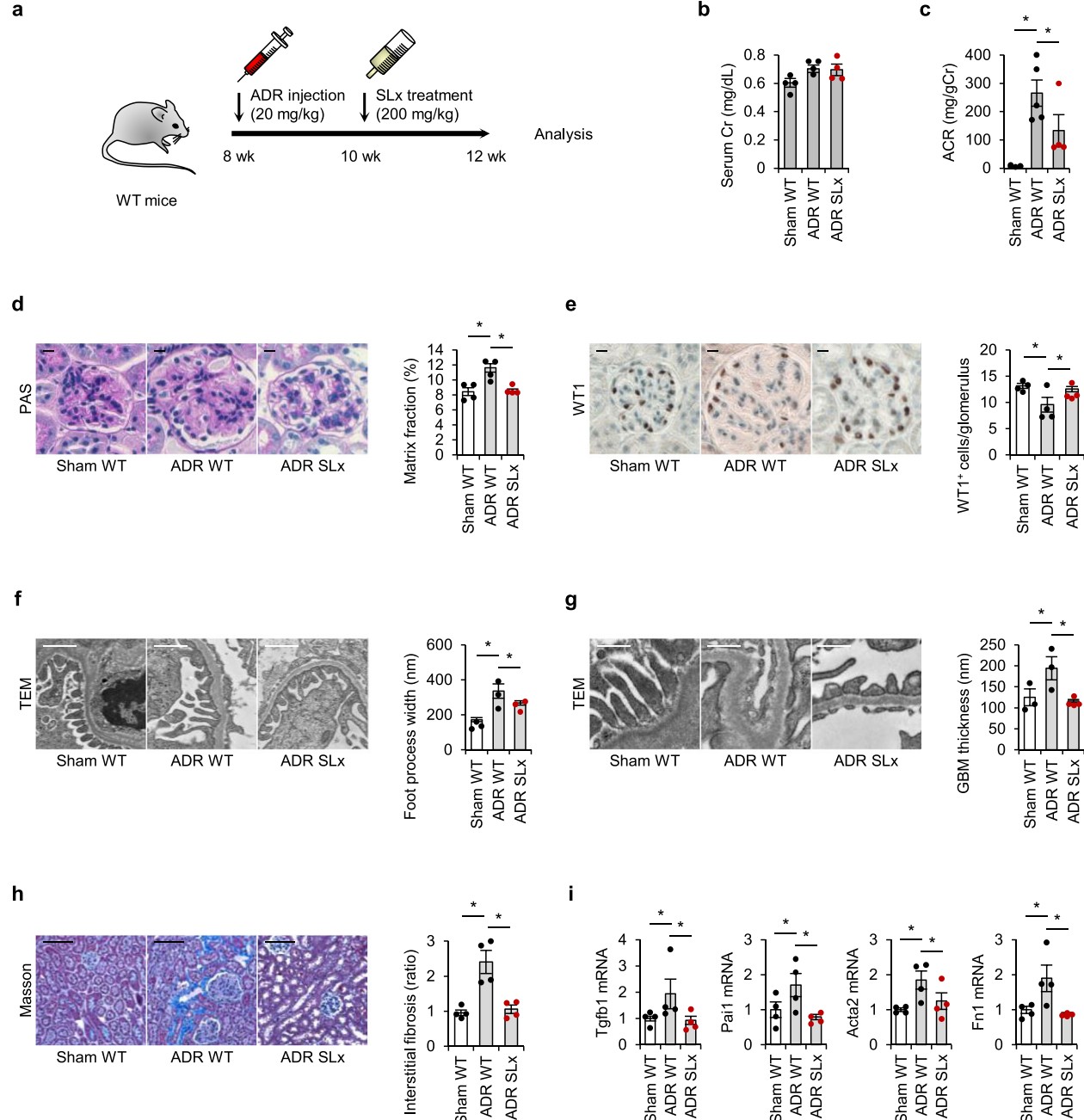

**Fig. 3 | Pharmacological inhibition of ROCK2 alleviates ADR-induced podocytopathy in mice. a** The intervention scheme that was used to treat ADR mice. **b** Serum Cr levels at 12 weeks of age ($n = 4$). **c** Urinary ACR in ADR-injected mice ($n = 4$-5). **d** Representative PAS-stained images of kidney glomeruli from mice. The scale bar on the top left represents 10 mm ($n = 4$). **e** WT1 immunostaining and quantification of WT1-positive cells in glomeruli from mice. The scale bar on the top left represents 10 μm ($n = 4$). **f** Foot process width examined by TEM. The scale bar on the top left represents 1 μm ($n = 3, 4$). **g** GBM thickness assessed by TEM. The scale bar on the top left represents 0.5 μm ($n = 3, 4$). **h** Masson's trichrome-stained kidney sections. The scale bar on the top left represents 100 μm. Fibrosis volume was quantified as the area stained positive for collagen ($n = 4$). **i** The transcript levels of fibrotic markers in the kidney ($n = 4$). *$p < 0.05$. Data represent the mean ± s.e.m.

function studies, we demonstrated the crucial roles of ROCK2 in podocyte health in several proteinuric kidney disease models (i.e. diabetic nephropathy, obesity-induced glomerulopathy). In addition, ROCK2 inhibition has been shown to be protective against tubulointerstitial fibrosis in a unilateral ureteral obstruction model[14]. When considered alongside these previous observations, the current work implicates ROCK2 as a key molecule for the development of a wide range of kidney diseases and thus adds important public health-related findings.

Given the importance of ROCK2 in the podocyte function, elucidating the upstream stimuli governing its gene expression and activation is of

particular interest. Our data indicate that ROCK2 gene expression is upregulated in human and murine models of FSGS. However, the mechanism regulating the elevation of ROCK2 in FSGS remains unclear. Several circulating factors have been considered pathogenic feed-forward enhancers of ROCK2 in FSGS because these have been detected in the sera of patients and experimental models of FSGS[15]. Among these, Hiroki et al. demonstrated that inflammatory stimuli (e.g., angiotensin II, interleukins) increase ROCK2 expression at both mRNA and protein levels[16].

Inflammatory cytokines and transforming growth factor β have been shown to upregulate ROCK2 function as evaluated by the extent of a

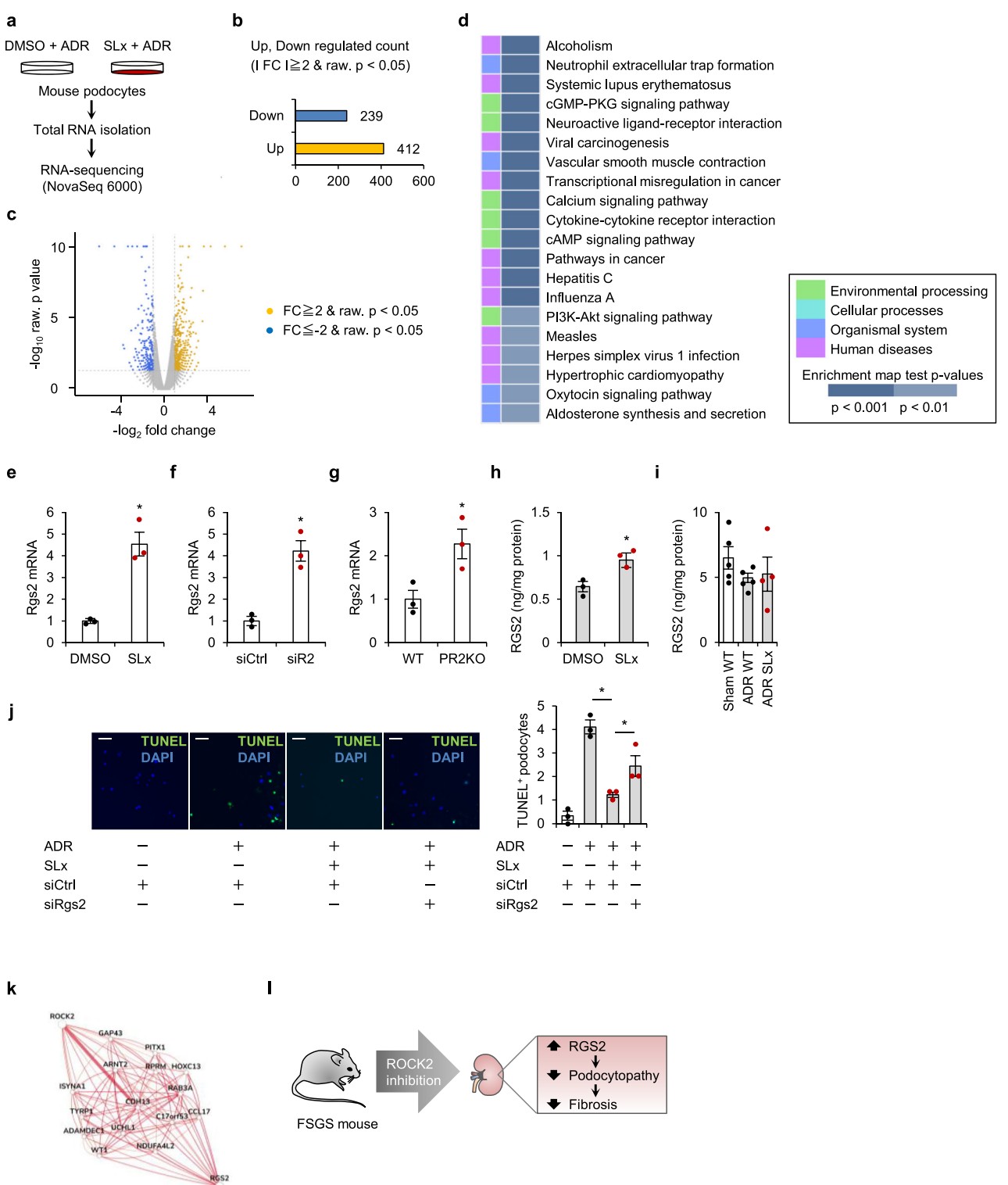

**Fig. 4 | ROCK2 modulates the cGMP pathway. a** The experimental protocol to prepare samples for high throughput RNA-sequencing. Cultured podocytes were treated with vehicle (DMSO) or ROCK2 inhibitor (SLx-2119). The number of regulated genes (**b**) and a volcano plot (**c**) of differentially expressed genes in podocytes based on log (fold change) > 2 with adjusted p < 0.05. **d** Results of a gene-set enrichment analysis (*p* value top 20). FC, fold change. **e** Rgs2 expression levels in podocytes treated with vehicle (DMSO) or SLx-2119 (*n* = 3). **f** Podocytes were transfected with control siRNA (siCtrl) or ROCK2 siRNA (siR2) to investigate Rgs2 mRNA levels (*n* = 3). **g** Rgs2 mRNA levels in primary podocytes (*n* = 3). **h** RGS2 prote*in* levels in ADR-stimulated podocytes. **i** RGS2 protein levels in the kidney of SLx-treated mice (*n* = 4, 5). **j** Representative microphotographs and the quantification of TUNEL-positive apoptotic podocytes. Podocytes were pretreated with siRNA against RGS2 before stimulation with ADR. The scale bar on the top left represents 50 μm (*n* = 3). **k** Functional network demonstrating the relationship between ROCK2 and RGS2 in podocytes (https://hb.flatironinstitute.org/; maximum number of genes, 15). **l** A schematic summary of key findings. In the setting of FSGS, podocyte ROCK2 is upregulated with proteinuria. ROCK2 inhibition prevents the loss of podocytes by recovering RGS2 signaling. *p* < 0.05. Data represent the mean ± s.e.m.

substrate of ROCK2[17]. In addition, ROCK2 can be activated by cleavage of the auto-inhibitory C-terminus, mediated by caspase 2 or granzyme B[18,19]. Of note, ROCK2 expression and activity were found to be related[16] and regulated by the peripheral clock gene BMAL1[20]. BMAL1 directly binds to the promoter of ROCK2 in a time-of-day-dependent manner, thereby modulating the time-of-day variation in ROCK2 activity; however, evidence is scant, so further research is needed.

The breadth of our approach, including the analysis of podocyte-specific ROCK2 deletion models in the context of ADR injury, is a strength of this study, which yielded several clear findings. We found ROCK2 signaling inhibition to be sufficient to prevent or attenuate podocyte loss, along with interstitial fibrosis. However, several studies have provided compelling evidence supporting the importance of the ROCK1 isoform in podocytes under physiological or pathological conditions. For instance, ROCK1 has been shown to regulate mitochondrial dynamics in podocytes[8]. Previous work from our group has shown that ROCK1 regulates AMP-activated protein kinase-mediated fatty acid metabolism. These studies indicate essential roles of ROCK in mediating the kidney function and structure; however, the fundamental question is whether or not inhibition of both ROCK isoforms provides additional benefits for proteinuric kidney disease. Future studies aimed at elucidating the interdependency between ROCK1 and ROCK2, through the generation of compound mutant models, will therefore prove beneficial.

The present findings will further expand our current understanding of the role of ROCK2 in podocyte biology. Several attempts have been made to explain the evolution from ROCK2 activation to progressive podocyte injury. In this study, we proposed that the molecular mechanism behind the effect of ROCK2 inhibition on the regulation of podocyte injury is most likely related to its ability to modulate cyclic nucleotide signaling, a mechanism implicated in the pathogenesis of kidney fibrosis. Our mechanistic investigations exploring the ROCK2-cyclic nucleotide signaling axis are in agreement with a recent observation showing antifibrotic effects of cGMP activation in unilateral ureteral obstruction models[21]. In addition, a large body of literature has reported drugs that elevate the cellular concentration of cGMP to inhibit kidney fibrosis. One gene that stood out as selectively upregulated in podocytes treated with ROCK2 inhibitor was RGS2, which is activated by PKG. Consistent with these findings, RGS2 deficiency has been demonstrated to accelerate kidney inflammation and fibrosis in unilateral ureteral obstruction models[11]. The GIANT indicated CDH13 as a mediator of ROCK2-dependent regulation of RGS2; however, whether or not the observed mechanism is due to the direct actions of ROCK2 remains unclear.

The present study showed that glomerular podocytes highly expressed ROCK2 in the setting of FSGS. Furthermore, we found that the genetic deletion of ROCK2 in podocytes prevented histological and functional abnormalities in FSGS mice. This finding prompted us to test whether or not pharmacological intervention for ROCK2 could attenuate kidney abnormalities seen in FSGS. Chemical inhibition of ROCK2 rescued podocyte damage and in turn kidney fibrosis. In addition to the known beneficial effects of ROCK2 inhibition in diabetic nephropathy and a unilateral ureteral obstruction model[14,17], this study identified novel effects on FSGS. Experiments using selective ROCK2 inhibitors in rodent models of kidney disease generally have preventive actions on the disease process, including reducing albuminuria and mesangial sclerosis, as well as decreasing GBM thickness. Thus, ROCK2 appeared to be a comprehensive therapeutic target for preventing or curing an abnormal kidney function, regardless of the etiology.

With respect to translating experimental findings into clinical trials, the therapeutic benefits of ROCK2 inhibition must be carefully weighed against the potential risk of toxicity. However, it should be noted that the safety and feasibility of ROCK2 inhibition in humans has already been established with belumosudil, an orally available selective ROCK2 inhibitor, in patients with chronic graft-versus-host disease[22]. In addition, clinical trials with ROCK2 inhibitors targeting diffuse cutaneous systemic sclerosis are ongoing[23]. Further trials involving ROCK2 inhibitors with the goal of reducing the risk

of an abnormal kidney function and death would engender additional confidence.

We acknowledge the limitation of the mouse background used in this study. In general, C57/BL6 mice are considered resistant to ADR-induced kidney injury as indicated by serum creatinine levels in our study. The serum levels of creatinine tended to be increased by ADR injection, but this increase was not statistically significant, which is consistent with a previous report[24]. An additional issue we could not dissect was the biological significance of ROCK2 as a regulator of chronic kidney disease due to other causes, such as hypertension, immune dysregulation, and others (e.g. nonproteinuric kidney disease). Because ROCK2 is expressed in a broad range of kidney cells, further studies using other models of glomerular and tubular disease are warranted. In addition, how podocyte damage evolves into progressive glomerular diseases remains unclear. These limitations should be considered when interpreting the findings of this study.

In conclusion, the present study demonstrates that podocyte ROCK2 is activated in the context of FSGS. Furthermore, genetic and pharmacological inhibition of ROCK2 significantly attenuated albuminuria and histological abnormalities in ADR-induced nephropathy. Our work sheds light on the molecular regulation of podocyte damage and provides a more complete understanding of ROCK2 as an orchestrator of kidney homeostasis.

## Methods

### Mice

Podocyte-specific ROCK2 knockout (PR2KO) mice were created on C57BL/6 background by mating the Rock2^flox/flox line with Nphs2-Cre mice obtained from The Jackson Laboratory. Rock2^flox/flox mice were generated by transgenic insertion of the LoxP site flanking exon 3 of the Rock2 gene and maintained at our facilities. Mice were kept in a temperature-controlled facility at 22 °C on a daily 12-h light/dark schedule and fed tap water and standard chow *ad libitum*.

In the first set of experiments, a single dose of 20 mg/kg doxorubicin hydrochloride (#D1515; Merck KGaA, Darmstadt, Germany) or vehicle (0.9% NaCl) was injected into the tail vein of 8-week-old male wild-type (WT) or PR2KO mice to induce ADR nephropathy. It has been suggested that C57BL mice are resistant to ADR-induced kidney damage but tissue injury is inducible at higher doses (13–25 mg/kg)[25–27] than those required in BALB/c mice. In the second set of experiments, 6-week-old male C57BL/6J mice were randomly divided into 3 experimental groups as follows: (1) vehicle; (2) ADR + vehicle; and (3) ADR + SLx-2119. ADR dissolved in 0.9% NaCl was injected intravenously once 2 weeks before SLx-2119 treatment. SLx-2119 (100 mg/kg) was administered every 12 h via orogastric gavage for 2 weeks. Significant inhibition of kidney ROCK2 activity was confirmed at this concentration in mice as previously demonstrated[17]. At 12 weeks old, serum and urine samples were collected from individual mice under isoflurane anesthesia. Kidney tissues were snap-frozen in liquid nitrogen and stored at –80 °C. The mice were observed at least once daily for signs of illness. Any signs of illness including lethargy, rapid breathing, skin discoloration, and paresis were reported in this study.

Studies dealing with animal use were conducted under protocols approved by the Committee on Ethical Animal Care and Use of the Jikei University School of Medicine.

### Histology

Kidney samples obtained from mice were fixed overnight in 10% buffered formalin and embedded in paraffin. Sections 3 μm in thickness were stained with periodic acid-Schiff for microscopic evaluations. Glomerular and mesangial areas were analyzed in 20 glomeruli per section. The area was quantified by the ImageJ software program (National Institutes of Health, Bethesda, MD). Kidney tissues were processed for Masson's trichrome staining to detect the levels of collagen deposition in the kidneys. Photographs were taken in 10 randomly selected areas with an EVOS M5000 Imaging System (Invitrogen, Waltham, MA). The extent of interstitial and perivascular fibrosis in kidney sections was quantified by the ImageJ

software program. Data are expressed as the ratio of stained area per total tissue area.

For immunofluorescence, 3-μm-thick paraffin-embedded sections were deparaffinized and subjected to antigen retrieval in citrate buffer. The sections were stained with an anti-ROCK2 antibody (#ab71598; Abcam, Cambridge, UK) and an anti-nephrin antibody (#BP5030; OriGene, Rockville, MD). To determine the loss of glomerular podocytes, the sections were stained with Wilms' tumor 1 (WT1) antibody (#ab89901; Abcam), and the number of positive cells was counted at least in 20 glomeruli for each mouse.

### Cell culture
No cell lines used in this study were found in the database of commonly misidentified cell lines that is maintained by ICLAC and NCBI BioSample. A conditionally immortalized murine podocyte cell line (E11) was obtained from Cell Line Services, but was not further authenticated after purchase. Mycoplasma-free E11 podocytes were propagated in 10 U/mL murine interferon-γ at 33 °C and then differentiated by culture for 10-14 days at 37 °C in the absence of interferon-γ[28]. ROCK2 knockdown podocytes were established by incubating cells with siRNA as described previously[9]. ROCK2 deletion did not affect the expression levels of ROCK1 in podocytes (Supplementary Fig. 2).

### Glomerular isolation
Mouse glomeruli were isolated as described elsewhere[17]. In brief, mice were perfused through the heart with magnetic Dynabeads (Invitrogen). After perfusion, the kidneys were removed, minced into small pieces, and digested by collagenase A in Hanks' balanced salt solution buffer. The digested tissue was then filtered through 100-μm and 70-μm cell strainers. Glomeruli containing Dynabeads were collected using a magnet.

### Primary culture of murine podocytes
Podocyte isolation was performed as previously described[28]. After removing erythrocytes with ammonium chloride potassium lysis buffer and depleting endothelial cells with CD31 antibody (#102504; Biolegend, San Diego, CA, USA), nephrin-positive cells were isolated from minced mouse kidneys using magnet-activated cell sorting with nephrin antibody (#PA5-25932; Thermo Fisher Scientific, Waltham, MA, USA).

### RNA isolation, quantitative real-time polymerase chain reaction (PCR), and RNA sequencing
Total RNA was prepared from kidney tissues and cultured podocytes using TRIzol reagent (Invitrogen). Next, 1 μg of total RNA was reverse-transcribed using the iScript RT Reagent Kit (Bio-Rad, Hercules, CA). To analyze the mRNA expression, real-time quantitative PCR was performed using the Thermal Cycler Dice Real Time System TP800 (Takara Bio, Shiga, Japan) with SYBR Green I fluorescence signals. The transcript levels of genes were normalized to β-actin and expressed as levels relative to the control. The primer sequences utilized for PCR are presented in Supplemental Table 1. RNA sequencing was performed by the Illumina NovaSeq6000 platform by Macrogen (Seoul, Korea).

### Protein analyses
ROCK2 protein abundance was detected by Western blot analyses as described previously[9]. The primary antibodies used were anti-ROCK2 antibody (#ab71598; Abcam) and anti-β-actin antibody (#sc-47778; Santa Cruz Biotechnology). The molecular size estimation was performed by referencing size markers present on the membrane during the experiment. RGS2 protein levels were measured using ELISA kit (#abx542128; Abbexa, Cambridge, UK).

### Transmission electron microscopy (TEM)
Kidney samples were fixed with 2% glutaraldehyde in 0.1 M phosphate buffer overnight at 4 °C and processed at the Electron Microscopy Facility at The Jikei University School of Medicine. The specimens were then postfixed with 1% osmium tetroxide in the same buffer at 4 °C for 2 h. Dehydration was performed using a graded series of ethanol washes, and then the sample was placed in propylene oxide and embedded in Epok 812 (Oken, Tokyo, Japan). Ultrathin sections were prepared with a diamond knife, and stained with uranium acetate and lead citrate. The sections were analyzed by a pathologist with a JEM-1400 Plus transmission electron microscope (JEOL, Tokyo, Japan) at 100 kV. Foot process width and glomerular basement membrane (GBM) thickness were examined in 20 positions of each mouse using the ImageJ software program. The foot process was defined as any connected epithelial segment butting on GBM between two neighboring filtration pores or slits. The GBM thickness was determined as the distance between the outer limit of the endothelium and the cell membrane of the podocyte foot process.

### Statistics and reproducibility
For each animal, different investigators were involved. An investigator administered ADR or SLx-2119. This investigator was the only person aware of the treatment group allocation. Other investigators were unaware of the treatment. Confounders were not controlled as experiments were processed randomly and individually. The sample size was determined based on the literature[2]. No data were excluded from the analysis. All experiments were performed in at least three biological replicates (see figure legends), and results are represented as the mean ± standard error of the mean (n as indicated in the figure legends). Statistical evaluations of two groups were performed by a two-tailed Student's $t$-test. Data involving more than two groups were evaluated by an analysis of variance and Bonferroni's post hoc correction. Pearson's correlation was used to analyze the associations between ROCK2 levels and urinary protein excretion. A value of $p < 0.05$ was considered to be statistically significant.

### Reporting summary
Further information on research design is available in the Nature Portfolio Reporting Summary linked to this article.

## Data availability
Numerical source data for graphs in the manuscript can be found in Supplementary Data 1 file. RNA sequencing data are available at NCBI database under GSE262013.

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

## Acknowledgements

This work was supported by Grant-in-Aid for Scientific Research from the Japan Society for the Promotion of Science (23K07709, 22K08347, 21K20914, 20K08645); the Astellas Foundation for Research on Metabolic Disorders; the Japan Diabetes Foundation; the Mochida Memorial Foundation for Medical and the Pharmaceutical Research; and the Ichiro Kanehara Foundation. We thank Yuko Niikura and Yuki Takemura for their excellent technical assistance with the electronic microscopy.

## Author contributions

K.M. designed and performed the research, analyzed the data, and wrote the manuscript. Y.N. helped with glomerular isolation. K.S., S.O. and E.M. assisted DNA genotyping. M.S. and T.T. helped with the histological examinations. D.K., K.U., Y.T. and R.N. wrote the manuscript. All authors read and commented on the manuscript.

## Competing interests

The authors declare no competing financial interests.
