## [Peer review file · Communications Biology]

Reviewers' comments:

Reviewer #1 (Remarks to the Author):

In this manuscript the authors showed increased expression of ROCK2 in human FSGS and in the adriamycin mouse model, with podocytes one of the cell types affected. Podocyte injury is important to the pathogenesis of FSGS. Deletion of ROCK2 in podocytes improved adriamycin injury, as did the ROCK2 inhibitor SLx-2119. Transcriptomic profiles were assessed in response to SLx in cultured podocytes, showing upregulation of genes involved in cyclic nucleotide signaling. This included *Rgs2* (regulator of G protein signaling 2), previously shown to prevent the progression of kidney fibrosis.

Comments:

The authors have previously generated podocyte-specific ROCK2 deleted mice and tested this in other models of kidney disease, somewhat lessening the novelty of this manuscript. However, assessment in other models is relevant to establishing generalizability of the role of ROCK2 in other etiologies of kidney fibrosis. In this manuscript, the use of both human and rodent *in vivo* data are a strength, as is the inclusion of cell culture studies in podocytes and RNAseq analysis to identify a potential mechanistic pathway. The data are generally well presented. However, addressing a number of points would help to clarify the data and strengthen some conclusions. These are outlined below.

1. The authors reference a previous paper showing reduced ROCK2 in podocytes - it would be valuable to include an image in this manuscript showing lack of ROCK2 in podocytes in the knockout model as the image included in the previous paper was not that convincing.
2. n=6-7 in Figure 2 and n=4-5 in Figure 3 are noted in the legend. However, the graphs show 2-4 data points. Given the small number of mice in the study, please include all of the mice for all of the assessments, and identify if a mouse has been left out and what the reason for this is.
3. For Figure 4, RNA seq was done for podocytes treated or not with the ROCK2 inhibitor. This does not really inform on the mechanism of protection in disease. It would be more meaningful to assess adriamycin-treated podocytes with and without the inhibitor. Subsequent studies would need to show that changes are also seen at the protein level, and in kidneys of SLx treated mice, to confirm relevance to pathology and *in vivo* effects. Mechanistic studies showing that increased expression (e.g. via transfection) of *Rgs2* protects against adriamycin injury similarly to SLx would also strengthen the importance of this pathway.
4. Please clarify the dosing of SLx-2119. The reference provided for the dosing describes giving 100mg/kg by gavage every 12 hours. Is this what was done?

Minor Comments:

- It would be helpful to point out what is being measured in the EM images in particular for a non-nephrology reader.
- The discussion in general is good and outlines the major points. However, I found paragraph 3 starting on line 278, page 11, to be somewhat confusing. Its discussion is a mixture of ROCK2 upregulation and activation - suggest making this clearer and perhaps splitting into two paragraphs to discuss the increase in expression vs an increase in activation.
- Has the increase in ROCK2 expression in glomeruli been shown to also result in increased ROCK2

activity?

Reviewer #2 (Remarks to the Author):

The authors demonstrated the role of ROCK2 in FSGS by using genetic and pharmacological inhibition of ROCKS in ADR-included mice FSGS model. The study designed generally well. But still, some concerns need to be addressed as following:

- 1, When inhibited ROCK2 by genetic or pharmacological way, does ROCK1 change?
- 2, The authors explored molecular mechanism by transcriptomic profiles assessing in SLx-2119-treated cultured podocytes. What should be noted is the widely side effect by drugs, so I strongly suggest the author using the PRKO primary podocyte to dig the molecular mechanism.
- 3, Pls. assess the role of RGS2 in mediating ROCK2's function, at least in ADR stimulated podocyte in vitro by using RGS2 siRNA or plasmid.
- 4, The quality of the morphology pictures including PAS, masson and WT1 staining should be strengthened.
- 5, Pls. also indicate the background of mice in the first set of experiments.

Responses to reviewers' comments

We would like to extend our gratitude to the editors and reviewers for their valuable comments and support regarding this work. We truly value your efforts and suggestions, which have significantly contributed to enhancing the quality of our work. Your recommendations have been thoroughly considered, and relevant adjustments have been incorporated. We sincerely hope that our revised manuscript meets your expectations, and we eagerly anticipate your feedback. The reviewer's comments are indicated in italics, followed by a detailed response addressing each point.

***Reviewer #1:** In this manuscript the authors showed increased expression of ROCK2 in human FSGS and in the adriamycin mouse model, with podocytes one of the cell types affected. Podocyte injury is important to the pathogenesis of FSGS. Deletion of ROCK2 in podocytes improved adriamycin injury, as did the ROCK2 inhibitor SLx-2119. Transcriptomic profiles were assessed in response to SLx in cultured podocytes, showing upregulation of genes involved in cyclic nucleotide signaling. This included Rgs2 (regulator of G protein signaling 2), previously shown to prevent the progression of kidney fibrosis. The authors have previously generated podocyte-specific ROCK2 deleted mice and tested this in other models of kidney disease, somewhat lessening the novelty of this manuscript. However, assessment in other models is relevant to establishing generalizability of the role of ROCK2 in other etiologies of kidney fibrosis. In this manuscript, the use of both human and rodent in vivo data are a strength, as is the inclusion of cell culture studies in podocytes and RNAseq analysis to identify a potential mechanistic pathway. The data are generally well presented. However, addressing a number of points would help to clarify the data and strengthen some conclusions. These are outlined below.*

Our response – We would like to thank the reviewer for the kind review and positive comments, as well as for the additional questions below, which helped to strengthen the manuscript.

Comment 1 – The authors reference a previous paper showing reduced ROCK2 in podocytes - it would be valuable to include an image in this manuscript showing lack of ROCK2 in podocytes in the knockout model as the image included in the previous paper was not that convincing.

Our response – We appreciate your feedback on our manuscript. Immunostaining of ROCK2 was performed in kidney tissues to demonstrate podocyte-specific deletion of ROCK2. Clear images have been added to Supplementary Fig. 1. We have revised our manuscript as follows (page 5),

To begin testing for functional links between ROCK2 expression and ADR-induced kidney damage, we crossed Rock2-flox mice with mice expressing Cre recombinase under the control of the Nphs2/Podocin promoter to generate mice with podocyte-specific disruption of ROCK2. Podocyte-specific ROCK2 ablation was confirmed by immunohistochemistry (Supplementary Fig. 1).

Comment 2 – n=6-7 in Figure 2 and n=4-5 in Figure 3 are noted in the legend. However, the graphs show 2-4 data points. Given the small number of mice in the study, please include all of the mice for all of the assessments, and identify if a mouse has been left out and what the reason for this is.

Our response – As noted by the reviewer, some of the presented graphs appear to contain only two samples. Although all data were analyzed and included in the figures, some values are so close that they are difficult to distinguish. To enhance the clarity of the data distribution, we have separated overlapping dots. Furthermore, we have uploaded all source data underlying the graphs as supplementary information for reference.

Comment 3 – For Figure 4, RNA seq was done for podocytes treated or not with the ROCK2 inhibitor. This does not really inform on the mechanism of protection in disease. It would be more meaningful to assess adriamycin-treated podocytes with and without the inhibitor. Subsequent studies would

need to show that changes are also seen at the protein level, and in kidneys of SLx treated mice, to confirm relevance to pathology and in vivo effects. Mechanistic studies showing that increased expression (e.g. via transfection) of Rgs2 protects against adriamycin injury similarly to SLx would also strengthen the importance of this pathway.

Our response – Thank you for your comment. As recommended, we conducted RNA-Seq analysis under adriamycin-stimulated conditions. As shown in Fig. 4d, the top differentially expressed pathways in podocytes included the cyclic GMP (cGMP)-dependent protein kinase (PKG) and cyclic AMP (cAMP) signaling pathways. The manuscript has been edited accordingly (page 7).

As shown in Fig. 4a, transcriptomic profiles were assessed in **ADR-stimulated podocytes**. The criteria for the detection of differentially expressed genes in this experiment were an absolute fold-change value of > 2 and high statistical significance (p value < 0.05). We found that **412** mRNAs were significantly upregulated, while **239** mRNAs were downregulated in ROCK2-inhibited podocytes (Figs. 4b and c).

Protein levels of RGS2 were examined by ELISA. As shown in Fig. 4h, RGS2 protein levels were increased by the treatment of SLx under ADR-stimulated conditions. However, similar results were not obtained in kidney tissue samples (Fig, 4i). These data indicate that ROCK2-mediated regulation of RGS2 is limited in podocytes. The result section has been edited as follows (page 7),

..... The protein levels of RGS2 were analyzed using ELISA. The results in Fig. 4h demonstrate that RGS2 protein levels increased with SLx treatment under ADR-stimulated conditions. However, kidney tissue samples did not yield similar results (Fig. 4i). These findings suggest that ROCK2-mediated regulation of RGS2 is restricted in podocytes.

As a mechanistic study, podocytes were treated with siRNA against RGS2 to assess the effect on ROCK2 function. We performed a TUNEL assay in podocytes to assess the role of RGS2. As shown in Fig. 4j, ADR-induced podocyte death was prevented by the treatment with SLx, a ROCK2 inhibitor. Importantly, the number of TUNEL-positive cells was increased by the RGS2 knockdown in ROCK2-inhibited podocytes. These data support the idea that the beneficial actions of ROCK2 inhibition are dependent, at least in part, on RGS2 upregulation. We have edited the manuscript as follows (page 7),

..... To assess the role of RGS2 in mediating ROCK2's function, we next conducted a TUNEL assay under ADR-stimulated conditions. As demonstrated in Fig. 4j, the number of TUNEL-positive cells was decreased by the treatment of SLx. However, the protective action of ROCK2 inhibition was partially canceled by the treatment of siRNA against RGS2. These data support the idea that the beneficial actions of ROCK2 inhibition are dependent, at least in part, on RGS2 upregulation.

Comment 4 – Please clarify the dosing of SLx-2119. The reference provided for the dosing describes giving 100mg/kg by gavage every 12 hours. Is this what was done?

Our response – SLx-2119 was administered every 12 h via orogastric gavage for 2 weeks at 100 mg/kg, the dose that we have confirmed decreased ROCK2 activity in the kidney (Am J Physiol Renal Physiol. 2019, PMID: 31364374). We have edited our manuscript as follows (page 8),

ADR dissolved in 0.9% NaCl was injected intravenously once 2 weeks before SLx-2119 treatment. SLx-2119 (100 mg/kg) was administered every 12 h via orogastric gavage for 2 weeks.

Comment 5 – It would be helpful to point out what is being measured in the EM images in particular

for a non-nephrology reader.

Our response – Thank you for your suggestion. We have edited the methods section to explain what was measured in the EM analysis. The manuscript was updated as follows (page 10),

Foot process width and glomerular basement membrane (GBM) thickness were examined in 20 positions of each mouse using the ImageJ software program. **The foot process was defined as any connected epithelial segment butting on GBM between two neighboring filtration pores or slits. The GBM thickness was determined as the distance between the outer limit of the endothelium and the cell membrane of the podocyte foot process.**

Comment 6 – The discussion in general is good and outlines the major points. However, I found paragraph 3 starting on line 278, page 11, to be somewhat confusing. Its discussion is a mixture of ROCK2 upregulation and activation - suggest making this clearer and perhaps splitting into two paragraphs to discuss the increase in expression vs an increase in activation.

Our response – Thank you for the constructive suggestions. We have improved this point in our revised manuscript and made additional modifications in response to your ideas. We have separated the paragraph into two parts to explain the regulatory mechanism of ROCK2 gene expression and kinase activity separately. Please see page 12 for the updated manuscript.

Given the importance of ROCK2 in the podocyte function, elucidating the upstream stimuli governing its gene expression and activation is of particular interest. Our data indicate that ROCK2 gene expression is upregulated in human and murine models of FSGS. However, the mechanism regulating the elevation of ROCK2 in FSGS remains unclear. Several circulating factors have been considered pathogenic feed-forward enhancers of ROCK2 in

FSGS because these have been detected in the sera of patients and experimental models of FSGS²⁰. Among these, Hiroki et al. demonstrated that inflammatory stimuli (e.g., angiotensin II, interleukins) increase ROCK2 expression at both mRNA and protein levels²¹.

Inflammatory cytokines and transforming growth factor β have been shown to upregulate ROCK2 function as evaluated by the extent of a substrate of ROCK2¹⁵. In addition, ROCK2 can be activated by cleavage of the auto-inhibitory C-terminus, mediated by caspase 2 or granzyme B^{22,23}. Of note, ROCK2 expression and activity were found to be related²¹ and regulated by the peripheral clock gene BMAL1²⁴. BMAL1 directly binds to the promoter of ROCK2 in a time-of-day-dependent manner, thereby modulating the time-of-day variation in ROCK2 activity; however, evidence is scant, so further research is needed.

Comment 7 – Has the increase in ROCK2 expression in glomeruli been shown to also result in increased ROCK2 activity?

Our response – We understand the reviewer's concern. It has been demonstrated that ROCK2 mRNA expression induced by angiotensin II resulted in an increase in the protein expression levels and kinase activity of ROCK2, as evidenced by the increased phosphorylation levels of a ROCK2 substrate (J Mol Cell Cardiol. 2004, PMID: 15276023). We have previously demonstrated that glomerular ROCK2 expression is increased in rodent models and patients with diabetic nephropathy (Commun Biol. 2022, PMID 35396346). In these settings, myosin phosphatase targeting subunit 1, a ROCK2 substrate, is phosphorylated in the kidney, suggesting increased ROCK2 activity (Front Pharmacol. 2021, PMID: 34557101). The clock gene BMAL1 has been suggested as a regulator of gene expression and kinase activity of ROCK2 (J Clin Invest. 2015, PMID: 25485682). The discussion part has been edited as follows (page 12),

Inflammatory cytokines and transforming growth factor β have been shown to upregulate ROCK2 function as evaluated by the extent of a substrate of ROCK2¹⁵. In addition,

ROCK2 can be activated by cleavage of the auto-inhibitory C-terminus, mediated by caspase 2 or granzyme B^{22,23}. Of note, ROCK2 expression and activity were found to be related²¹ and regulated by the peripheral clock gene BMAL1²⁴. BMAL1 directly binds to the promoter of ROCK2 in a time-of-day-dependent manner, thereby modulating the time-of-day variation in ROCK2 activity; however, evidence is scant, so further research is needed.

Reviewer #2: *The authors demonstrated the role of ROCK2 in FSGS by using genetic and pharmacological inhibition of ROCKS in ADR-included mice FSGS model. The study designed generally well. But still, some concerns need to be addressed as following:*

Our response – We thank the referee for these insightful comments and constructive suggestions.

Comment 1 – When inhibited ROCK2 by genetic or pharmacological way, does ROCK1 change?

Our response - We appreciate your thoughtful criticism of our manuscript. Recent studies have shown no compensatory change in ROCK1 of heterozygous ROCK2 deficient mice (Eur Respir J. 2011, PMID: 21565918). Similar results were obtained by other investigators (J Clin Invest. 2008, PMID: 18414683, Heliyon. 2023, PMID: 36950615). To confirm the effects of ROCK2 inhibition on ROCK1 levels in podocytes, we performed qPCR analysis in podocytes treated with siRNA against ROCK2. As shown in Supplementary Fig. 2, ROCK1 expression levels were not changed in ROCK2-deficient podocytes. Taken together, these data indicate that compensatory change in the expression of the ROCK1 isoform does not occur in response to the loss of the ROCK2 isoform as previously reported in other cell types. We have edited our method section as follows (page 9),

..... ROCK2 knockdown podocytes were established by incubating cells with siRNA as described previously⁹. ROCK2 deletion did not affect the expression levels of ROCK1 in

podocytes (Supplementary Fig. 2).

Comment 2 – The authors explored molecular mechanism by transcriptomic profiles assessing in SLx-2119-treated cultured podocytes. What should be noted is the widely side effect by drugs, so I strongly suggest the author using the PRKO primary podocyte to dig the molecular mechanism.

Our response - Thank you for the constructive suggestions. As recommended by the reviewer, we have isolated podocytes from the kidney tissues of PR2KO mice. Rgs2 was upregulated in primary podocytes obtained from PR2KO (Fig. 4g). We have edited the manuscript as follows (page 7).

..... Induction of RGS2 was confirmed by quantitative PCR in podocytes treated with ROCK2 inhibitor (Fig. 4e), siRNA against ROCK2 (Fig. 4f), and primary podocytes obtained from PR2KO mice (Fig. 4g).

Comment 3 – Pls. assess the role of RGS2 in mediating ROCK2's function, at least in ADR stimulated podocyte in vitro by using RGS2 siRNA or plasmid.

Our response - We thank the reviewer for raising the important issue on the effects of ROCK2 deletion. Based on the reviewer's suggestion, we have conducted experiments to determine if ROCK2 inhibitor-mediated cytoprotective effects are dependent on RGS2. As shown in revised Fig. 4j, the protective action of ROCK2 inhibition was significantly attenuated by the treatment of siRNA against RGS2. However, this was partial cancelation, indicating that ROCK2's function is not completely dependent on RGS2. Other biological pathways may also be regulated by ROCK2 inhibition. We have edited the manuscript as follows (page 7).

..... To assess the role of RGS2 in mediating ROCK2's function, we conducted a TUNEL assay under ADR-stimulated conditions. As demonstrated in Fig. 4j, the number of TUNEL-

positive cells was decreased by the treatment of SLx. However, the protective action of ROCK2 inhibition was partially canceled by the treatment of siRNA against RGS2. These data support the idea that the beneficial actions of ROCK2 inhibition are dependent, at least in part, on RGS2 upregulation.

Comment 4 – The quality of the morphology pictures including PAS, masson and WT1 staining should be strengthened.

Our response – We have submitted separate high resolution figure files for your reference.

Comment 5 – Pls. also indicate the background of mice in the first set of experiments.

Our response – Thank you for pointing that out. Our mice were on a C57BL/6 background. Methods section was edited as follows (page 7),

Podocyte-specific ROCK2 knockout (PR2KO) mice were created on C57BL/6 background by mating the Rock2^{flox/flox} line with Nphs2-Cre mice obtained from The Jackson Laboratory.

.....

REVIEWERS' COMMENTS:

Reviewer #1 (Remarks to the Author):

The authors have responded thoroughly and thoughtfully to all of the comments, with significant improvement in the manuscript. One minor wording change is suggested for clarity: on page 7 lines 170-171, suggest changing this to "These findings suggest that ROCK2-mediated regulation of RGS2 may be restricted to podocytes" rather than "is restricted in podocytes".

Reviewer #2 (Remarks to the Author):

All of my concerns and questions have been adequately addressed.